# Impact of Treating Hatching Eggs with Curcumin after Exposure to Thermal Stress on Embryonic Development, Hatchability, Physiological Body Reactions, and Hormonal Profiles of Dokki-4 Chickens

**DOI:** 10.3390/ani11113220

**Published:** 2021-11-11

**Authors:** Ahmed Abdel-Kareem Abuoghaba, Mona A. Ragab, Soheir A. Shazly, Dariusz Kokoszyński, Mohamed Saleh

**Affiliations:** 1Department of Poultry Production, Faculty of Agriculture, Sohag University, Sohag 82524, Egypt; Abuoghaba@yahoo.com (A.A.-K.A.); moh_sa_al@yahoo.com (M.S.); 2Agricultural Research Center, Institute of Animal Production Research, Ministry of Agriculture, Dokki, Giza 12618, Egypt; drmonaragab111@gmail.com (M.A.R.); sohershazley@gmail.com (S.A.S.); 3Department of Animal Breeding and Nutrition, Faculty of Animal Breeding and Biology, Bydgoszcz University of Science and Technology, 85084 Bydgoszcz, Poland

**Keywords:** incubation temperature, curcumin, embryonic development, hatchability, T3 hormone

## Abstract

**Simple Summary:**

Curcumin has been used as a suitable feed supplement for poultry to improve several hematological and biochemical indicators, diminish heat stress, and increase antioxidant activity. This experiment evaluated the effects of incubation temperatures and spraying hatching eggs with curcumin during the incubation phase on chick embryo development, hatchability, physiological body reactions, and hormonal profiles of Dokki 4 chickens. The findings indicated that the relative water loss and dead after piping in the chronic incubation temperature group were significantly increased compared with the normal incubation temperature group. Post-hatch, the triiodothyronine level in the high incubation temperature group was significantly decreased than in the control group. Regarding curcumin treatments, relative water loss from eggs and the hatchability of fertile eggs in treated groups significantly increased, while body surface temperature significantly decreased compared with the control.

**Abstract:**

This study evaluated the impact of incubation temperature and spraying hatching eggs with curcumin during the early embryogenesis phase on chick embryo developments, hatchability, physiological body reactions, and hormonal profiles of Dokki 4 chickens. A total of 720 fertile eggs were equally distributed into two groups. In the first group, the eggs were incubated at normal incubation temperature/NIT (37.8 °C and 55–60% RH) for up to 19 days of incubation, whereas those in the second group were incubated in the same conditions except from 6 to 8 day, in which they were daily exposed to chronic incubation temperature/CIT (39.0 °C) for 3 h. Each group was classified into four curcumin treatment doses; the 1st treatment (control) was sprayed with distilled water, while the 2nd, 3rd, and 4th treatments were sprayed with 250, 500, and 1000 mg curcumin/liter distilled water. The results indicated that the lowest hatchability of fertile eggs (%) was obtained in the CIT group (*p* = 0.02), whereas the highest body surface temperature/BST compared in the NIT group (*p* = 0.01). Regarding curcumin treatments, the percentages of heart, gizzard, spleen, and T3 hormone levels in the treated group were significantly increased, while the H/L ratio was significantly reduced (*p* = 0.001) compared with the control. At 8 weeks of age, the testes and ovary percentages in treated groups were significantly (*p* = 0.05) increased compared with the control. In conclusion, exposure of hatching eggs to high thermal stress (39 °C) during the incubation phase had deleterious effects on chick performance and T3 hormone level. Moreover, spraying hatching eggs had beneficial impacts on growth, reproductive organs, T3 hormone level, and reducing H/L ratio.

## 1. Introduction

During the incubation phase, the eggshell temperature controls chick embryo development, hatchability, and chick performance [1]. Therefore, high embryo growth rates during the incubation phase seem to be sensitive to temperature fluctuations [2]. The chicken’s optimum incubation temperature for obtaining the highest hatchability, chick quality, and longest chick length at hatch, up to day 19 is 37.8 °C [3]. The eggs during the first incubation stage contain live embryos, which absorb heat from the interior incubator air, resulting in a lower embryo temperature than the incubator temperature [4,5]. The optimum incubation temperature ranges, especially after embryonic day 9, are difficult to reach due to excessive heat production from the developing embryos [3,6]. The increase in incubation temperature by about 0.5 or 1 °C (from 37.5 to 38.0 or 38.5 °C), especially from 3 to 25 days of the incubation phase, significantly affects turkey’s embryonic development [7]. Hence, controlling chick embryo temperature rather than incubator temperature remarkably improves chick quality [6]. The increase in incubation temperature above 37 °C harms broiler growth and the relative heart weight by 17–31% [3]. The findings of Leksrisompong et al. [4] revealed that an increased eggshell temperature higher than 39.5 °C retarded organ growth of embryos. Moreover, the embryos exposed to high incubation temperature suppressed crop, gizzard, liver, and intestine development [8], and it was found that the heart was the most consistently affected organ by high temperature [9]. The incubation temperature may affect the bird’s thermoregulation after hatching. Therefore, the peculiarity of epigenetic adaptation, which may occur during early pre-or postnatal ontogeny, may alleviate problems associated with the control of thermoregulation during rearing [10]. This reasoning suggests that modifying standard incubation temperatures by increasing broiler embryo temperatures may be a way to enhance the post-hatching performance of chickens [11]. 

Curcumin is used as a food additive in poultry feed due to its favorable effects as an antioxidant, antimicrobial, anti-inflammatory, and its immune-stimulant properties [12], orally in rats [13] as well as a preventative against oxidative stress [14]. The commercial curcumin extracts containing 75% curcumin, 20% dimethoxy-curcumin, and 5% bisdemethoxycurcumin, respectively, are thought to be biologically active and possess protective properties [15]. Studies on curcumin in the animal field are still in their infancy. Furthermore, curcumin ameliorates heat stress by modulating hepatic nuclear transcription factors and heat shock protein 70 in heat-stressed quails and broilers [16,17]. It was reported that turmeric powder supplementation in the diet improves immunological responses by increasing total immune globulin and IgG titers after injection with sheep red blood cells in laying hens [18]. Curcumin improves the endogenous secretion of the digestive enzymes in the broiler chicks [19] and reduces lipid peroxidation [20]. Curcumin is a potent inhibitor of histone deacetylases (HDACs), thus, it has an effect on the activities of HDACs and histone acetyltransferases (HATs) that regulate the acetylation and deacetylation of proteins involved in gene silencing and transcription [21]. Curcumin may control acetylation and deacetylation by reducing the adverse effect of oxidative stress [22]. Moreover, curcumin modulates DNA methylation through inhibiting DNA methyltransferases (DNMTs) [23]. DNA histone modifications and methylation are crucial epigenetic modifications of the genome that are involved in regulating embryonic development, therefore, curcumin can deactivate, suppress or reduce DNA methylation [24]. The injection technique for hatching eggs during the incubation period is to be considered a stressful procedure for embryo growing; in addition, it requires experience and skill for application, especially for small frames in developing countries [25]. Thus, temperature control is not affordable in developing countries; therefore, the aim was to determine if curcumin has an effect of reducing heat burden in the summer for poultry farms. 

A further aim of this experiment was to evaluate the impact of spraying hatching eggs after exposure to thermal stress (39 °C) with curcumin at 250, 500 and 1000 mg curcumin/liter distilled water, during early embryogenesis, on hatchability, chick quality, and organ development, performance, and some physiological body reactions of Dokki 4 chickens.

## 2. Materials and Methods

This study was performed in the Experimental Poultry Farm, Department of Poultry Production, Faculty of Agriculture, Sohag University, Egypt. This experiment was approved by the local Ethics Committee (6-1-2020).

### 2.1. Experimental Eggs and Management

In this study, 720 Dokki-4 fertile eggs (47 ± 2 g) were collected (3 times/day) from a hen breeder flock at 46 weeks of age. All eggs were equally distributed into two groups (2 groups × 4 treatment × 3 replicates × 30 eggs). In the first group (control), the eggs were incubated at 37.8 °C and 55–60 relative humidity up to 19 days of incubation phase, while those in the 2nd group were incubated at the same incubation conditions except for 3 days from the 6th to 8th day of incubation, in which eggs were daily exposed to 39.0 °C for 3 h from 12 to 3 p.m. in a separate incubator with the same relative humidity. Each group was classified into four treatment curcumin doses; the 1st one was spraying with distilled water (control), while the 2nd, 3rd, and 4th treatments were sprayed daily with 250, 500, and 1000 mg curcumin/liter distilled water. All eggs in the incubator were turned through 45° automatically each hour during the incubation phase.

### 2.2. Preparation of Curcumin Suspension and Egg Treatments

The curcumin suspension was diluted in Mili-Q distilled water. The distilled water and suspension were set at 37.8 °C for two hours in the incubator before spraying. Curcumin was purchased from Amazon stores, turmeric curcumin 100% pure extract 95% curcuminoids, made in the USA.

The eggshell surfaces were cleaned with 100% ethanol and then sprayed daily with 50 mL/1000 eggs for distilled water or curcumin suspension after heat stress (39.0 °C) exposure. Eggs were horizontally placed, and the suspension was applied opposite to the air cell of the egg surface. On the 6, 7, and 8th day of the incubation phase, eggs were transferred to another incubator and exposed to 39.0 °C and 55–60% RH to 3 h per day and then sprayed daily with curcumin solution. From 19 to 21 days, the eggs were exposed to 37.5 °C and 55–60% RH.

### 2.3. Studied Traits

#### 2.3.1. Relative Water Loss

Egg water loss was calculated as the difference in egg weight before placement and on day 8 of incubation and consequently expressed as a percentage of initial egg weight. 

#### 2.3.2. Hatchling Body Temperature and Chick Quality

At hatch, the hatchability was expressed as a percentage of fertile and set eggs [1]. Late-term embryonic mortality was determined by opened un-hatched eggs. Embryonic mortality values were calculated as follows: (number of dead embryos/all viable eggs at transfer) × 100%. Dead after piping was calculated as follows: (number of dead chicks after piping/number of eggs at transfer) × 100. Chick hatching weight for all chicks was determined by using a balance at ±0.1 g precision. At hatch, 72 chicks from each group (2 group × 4 treatments × 3 replicate × 3 chicks) were randomly taken to determine chick weight, chick length, and then chicks were euthanized killed to measure the weights of heart, gizzard, liver, intestine, and spleen. Chick length (cm) was measured from the beak tip of the middle toe [26]. The chick length and internal organs traits were measured approximately 12 h after hatch.

#### 2.3.3. Physiological Body Reactions

The head, wing, back, and shank temperatures of chickens were recorded by using an infrared thermometer, while the rectal temperatures (°C) were measured using a digital thermometer (to the nearest 0.1 °C) that was inserted into the rectal at 1 cm deep. The average body surface temperature (BST/°C) was calculated by the equation as described by Richard [27] as follow: BST/°C = (0.12 × wing T) + (0.03 × head T) + (0.15 × shank T) + (0.70 × back T). 

#### 2.3.4. Estimation of Hematological Variables, Glucose Level, and T3 Hormone Concentration

After 12 h from the hatch, newly chicks were slaughtered to collect blood samples, 72 blood samples (one ml each) were collected (2 groups × 4 treatment × 3 replicates × 3 chicks) were collected into heparinized tubules for hematological study. Blood samples were centrifuged, plasma separated, decanted, and deep-frozen for analysis. Hemoglobin (Hb) concentration was spectrophotometrically measured. Red blood cell (RBC/106) count was estimated using a hemocytometer. The granular and nongranular WBCs were based on the procedures of Gross and Siegel [28]. Briefly, one drop of blood was smeared on the glass slide. The smears were stained using Wright’s stain. One hundred leucocytes, including granular and nongranular, were counted on different microscopic fields representing 100 cells, and the heterophil to lymphocyte (H/L) ratio was calculated. Plasma triiodothyronine (T3) was determined by an enzyme-linked immunosorbent assay (ELISA) kit (International Reagents Corporation, Kobe, Japan). Plasma glucose concentration was measured using commercial kits [29].

#### 2.3.5. Productive Traits

A total of 192 chicks (2 groups × 4 treatment × 3 replicates × 8 chicks) were randomly allocated into two control and chronic (high temperature) groups. The chicks were placed in 24 battery cages to provide 8 chicks/replicate. The chicks were weighed using a balance at ±0.1 g precision on the first day of the growing period. The chicks received a standard pelleted broiler starter diet (22.5% CP and ME 12.8 MJ/kg of diet) between days 1 to 14, a grower diet (22.0% CP and ME 13.3 MJ/kg of diet) between days 15 to 28, and a finisher diet (21.0% CP and ME 13.5 MJ/kg of diet) between days 29 to 42 to exceed requirements by the NRC [30]. Feed and water were offered ad libitum during the growing period. The chicks were exposed to 23, 20, and 23 h of light during 0–7 d, 8–28 d and 29-end with (30–40 lx), (10–15 lx) and (3–5 lx) light intensity [31]. The mortality rate by cage was recorded daily during the trial.

#### 2.3.6. Carcass Characteristics

At 8 weeks of age, 96 chicks (2 groups × 4 treatment × 3 replicates × 4 chicks) were randomly taken and slaughtered to measure the body weight, carcass, liver, spleen, bursa, and gizzard, ovary, and testes. 

### 2.4. Statistical Analysis

The data were subjected to statistical analysis according to a general linear model (GLM) of program SAS [32] according to the following statically model: Yijk = μ + Ii + Cj + ICij + eijk where: Yijk, an observation; μ, overall mean; Ii, the effect of incubation temperature (i = 1, 2); Cj, the effect of the curcumin treatment (j = 1, 2, 3, 4); ICij, the effect of interaction between incubation temperature and curcumin treatment; and eij, random error. The significant differences between treatment means were determined by using Duncan [33]. The results were considered significantly different if *p* < 0.05 and tendencies were noted at *p*-values ≤ 0.05. An arcsine transformation was used for embryonic mortality to obtain normally distributed data, while hatchability percentages before comparison and analysis; were homogeneous and did not need a transformation to the arcsine angle.

## 3. Results

### 3.1. Relative Water Loss, Embryonic Mortality, and Hatchability

The effects of incubation temperatures, curcumin manipulations, and their interaction on relative water loss and embryonic mortality are presented in Table 1. The findings showed that the highest relative water loss and dead after piping for chick embryos were obtained in the chronic group (*p* = 0.004 and *p* = 0.006) compared with the control group. Regarding curcumin treatment, the relative water loss for eggs in treated groups was significantly (*p* = 0.037) decreased compared with a control group, while the dead after piping and embryonic mortality percentages were not affected. There was no significant interaction between incubation temperature and curcumin treatment for all studied traits. The findings revealed that the hatchability of fertile eggs was significantly (*p* = 0.019) decreased by incubation temperature, while the hatchability of set eggs was not affected. 

Regarding curcumin treatment, the results showed that the hatchability of fertile eggs was significantly (*p* = 0.033) increased, while the hatchability of set eggs was not affected. There was no significant interaction between incubation temperature and curcumin treatment for hatchability of fertile eggs and hatchability of set eggs.

### 3.2. Hatchling Quality Traits and Physiological Body Reactions

The impact of incubation temperature, curcumin manipulations, and their interaction on hatchling quality traits and physiological body reactions of newly-hatched chicks is shown in Table 2.

From the data presented in Table 2, chick weight at hatch (g) in the chronic group was significantly (*p* = 0.02) decreased, while relative chick weight (%) was not affected compared with a control group. The chick length was insignificantly higher in the control group, which amounted to 15.95 cm than 15.92 cm in the chronic group (*p* = 0.913). In addition, CWAH (g), RCW (%), and chick length (cm) were not influenced by curcumin treatments. There was no significant interaction between incubation temperature and curcumin treatment for CWAH (g), RCW (%), and chick length (cm).

The rectal temperature (RT/°C) and body surface temperature (BST/°C) for chicks produced from eggs exposed to high incubation temperature were significantly (*p* = 0.01) increased compared with those in the control group. CT (°C) and BST (°C) for chicks produced from eggs treated with curcumin were significantly (*p* = 0.01) decreased compared with the control group. There was no significant interaction between incubation temperature and curcumin treatment for CT (°C) and BST (°C).

### 3.3. Hatchling Organ Percentage

The effects of incubation temperatures, curcumin manipulations, and their interaction on the hatchling organ percentage are presented in
Table 3.

The heart and gizzard percentages for chicks in the chronic group were significantly (*p* = 0.01) lowered, while the intestine, liver, and spleen percentages were not affected compared to the control group. Regarding the effect of curcumin treatments, the percentages of the heart, gizzard, and spleen were significantly increased (*p* = 0.01), while the percentages of the intestine and liver were not affected compared with the control group. All hatchling organ percentages, except gizzard percentage, were not affected by the interaction.

### 3.4. Hematological Parameters, Glucose Level, and T3 Hormone Concentration

Hematological parameters, glucose level, and T3 hormone concentration in newly hatched chicks are presented in Table 4.

From the data in Table 4, the RBCs (×10^6^) and WBCs (×10^3^) were significantly increased by incubation temperature (*p* = 0.001), while hemoglobin (g dL^−1^) was not affected. The findings showed that the H/L ratio was significantly (*p* = 0.001) increased in chicks produced from eggs exposed to high temperatures during the incubation phase compared with the control group. Triiodothyronine hormone and glucose levels in the chronic group were significantly (*p* = 0.001) decreased compared with those of the control group.

Regarding the effect of curcumin treatments, the RBCs, WBCs, and Hb concentration was not affected, while the H/L ratio was significantly decreased (*p* < 0.001). The glucose and T3 levels in the treated groups significantly (*p* < 0.001) increased compared with the control group. The H/L ratio, glucose, and T3 levels were significantly (*p* < 0.001) affected by the interaction.

### 3.5. Productive Performance

Data presented in Table 5 shows that the effects of incubation temperatures, curcumin manipulations on the body weight, cumulative feed consumption, and feed conversion ratio up to 8 weeks of age. There was a significant (*p* < 0.01) decrease in the live body weights at one day, 2nd, 4th, 6th, and 8th week for chicks in the chronic group than that of the control group. The total body weight gain and daily weight gain for chicks in the chronic group were significantly (*p* < 0.01) lowered compared with those in the control group. The cumulative feed consumption up to 8 weeks of age was significantly (*p* < 0.01) increased in the control (1529.2 g) than (1436.7 g) in the chronic group. During the 8 weeks growing period, the chronic (2.33) group had an insignificantly higher FCR relative to the control group (2.25).

Regarding curcumin treatments, these findings indicated that the body weight at the 8th week of age, total body weight change, and weight gain for chicks in the treated group significantly (*p* = 0.001) increased compared with those in the control group. The highest total feed consumption was observed in the 4th treatment, which was significantly increased compared with those in the 1st, 2nd, 3rd groups. 

H/L, Heterophile/lymphocyte ratio. The values are the average of group data.

### 3.6. Internal Organ Percentages and Immune Organs

Findings in Table 6 and Table 7 indicate that the male slaughter weight (g) in the chronic group was significantly (*p* = 0.001) decreased, while the percentages of heart, intestine, gizzard, liver, spleen, testes, and eviscerated carcass were insignificantly decreased compared with the control group. The bursa of Fabricius percentage was significantly (*p* = 0.05) lowered in the chronic group (0.223) than that (0.246) in the control group. Regarding curcumin treatment, the slaughter weight (g), and percentages of the intestine, gizzard, liver, and testes were significantly (*p* < 0.05) increased by curcumin treatment compared with the control group. Referring to female chicks, the results in Table 6 show that the slaughter weight (g) and the percentages of intestine and bursa of Fabricius in the chronic group were significantly (*p* = 0.05) decreased than that in the control one. The slaughter weight (g), percentages of gizzard, and ovary in curcumin-treated chicks were significantly (*p* < 0.05) increased compared with the control group. 

## 4. Discussion

The significant increase of relative water loss at 8 days for eggs exposed to high temperature could be attributed to increased water evaporation from eggs and consequently decreased egg weight. These findings agreed with those of Walstra et al. [34], who found increasing incubation temperature daily to 40.7 °C for 3 h from 15 to 17 days of incubation, leading to a decrease in yolk sac weight. In addition, the embryos are more sensitive to high temperatures because the incubation temperatures above 37.5 °C promote excessive egg water loss [35].

The decrease in the relative water loss (%) for eggs treated with curcumin could be attributed to improving eggshell strength leading to a decrease in relative water loss from eggs. The improvement of eggshell quality of laying hens supplemented with curcumin at 100, 150, or 200 mg/kg diet is mainly due to the increased feed consumption, which frees calcium in the blood serum combined with plasma proteins to obtain enough Ca^2+^ in the blood to participate in the formation of the eggshell [36]. They also reported that eggshell thickness and eggshell strength produced from laying hens were significantly increased than the control group. They also added that the diet supplemented with 100 mg/kg of curcumin significantly (*p* > 0.05) increased albumen height by 21.12%. The obtained findings indicated that the dead after piping in the chronic group was significantly (*p* = 0.01) decreased compared with the control group. This reflects insufficient egg contents to chick embryo development due to exposed high incubation temperature. This may also be due to accelerated in-ovo development, as confirmed by their higher conductive, evaporative heat losses, higher conductance, and, consequently the higher metabolic rates [37]. The significant (*p* = 0.01) increase in the dead after piping for chick embryos in the chronic group could be attributed to the deleterious effects of heat stress, which negatively affect the pulmonary vascular capacity leading to increasing the metabolic oxygen demand [38].

The lowest hatchability of fertile eggs in the chronic group may be due to the increased water loss from eggs, and consequently, insufficient egg contents to chick embryo development. These results are in agreement with those of Abuoghaba [25], which indicated that the hatchability of fertile broiler eggs exposed to chronic incubation temperature (40 °C) group was significantly (*p <* 0.05) decreased compared with those control group. The increased hatchability of fertile eggs in the treated groups could be attributed to improving residual egg yolk content, and consequently, improved hatchability. The improved hatchability percentage of fertile eggs may be due to the ability of curcumin to interact within the eggshell cuticle, which allows chick embryos to break eggshell at hatch. Also, the improved hatchability in the groups treated could also be attributed to the active compounds in curcumin promoting the proliferation of epithelial and tubular gland cells in the magnum to albumen synthesis and secretion [25].

The increased chick weights at the hatch for chicks produced from eggs in the control group may be due to the increased yolk sac uptake into the embryo abdomen, which provides chick nutrients requirements during the first few days of life [6]. These results agreed with those of Abuoghaba [25], who found that the higher chick was obtained in the control, while the lower values were recorded in the chronic high group at 12 h after hatch.

The increased rectal and body surface temperatures may be due to the adverse effect of high incubation temperature, which led to an increase in the chick’s physiological body reactions. These findings agreed with those of Abuoghaba [25], who found that the rectal and body surface temperatures were significantly increased with increasing incubation temperatures.

The decreased rectal and body surface temperatures for chicks produced from eggs treated with curcumin may be due to improving physiological body status and antioxidant capacity for treated chicks compared with control groups. The finding disagreed with those of Oke [39], who found no difference in the rectal temperatures of the birds fed diets supplemented with turmeric powder for 8 weeks.

The percentages of heart and gizzard for chicks at the hatch were significantly decreased in the chronic group compared with the control. The significant decrease in the heart for chicks in the chronic group may be attributed to the higher vulnerability and occurrence of metabolic disorders related to cardiovascular development, which reflects the considerable fall in heart percentage for chickens in the chronic group [4]. The incubation temperature manipulation resulted in heart hypoplasia, indicating that high incubation temperatures may affect heart weight. Therefore, heart hypoplasia reduces oxygen supply to the tissues and may consequently impair body development [37]. The results of Molenaar et al. [1] indicated that broilers produced from eggs incubated at high temperatures presented a higher incidence of ascites during the growing period compared with those incubated at normal temperatures.

In this study, the significant increase of heart, gizzard, and spleen in the chicks produced from eggs treated with curcumin could be attributed to improved antioxidant activity by increasing the SOD and GSH-Px activity, and decreasing MDA levels. Also, this may be attributed to curcumin’s ability to decrease protein oxidation levels in the organs and minimize the cellular fluid loss in newly chicks after exposure to thermal stress.

These findings agreed with those of Rajput et al. [40], who found that dietary curcumin supplementation significantly improved the dressing percentage and breast yield in heat-stressed broiler chickens. They attributed this to the ability of dietary curcumin to decrease the level of protein oxidation in the breast muscles and minimize the loss of cellular fluids in stressed broiler chickens.

The findings of Sahin et al. [16] showed that curcumin could alleviate oxidative stress by modifying the hepatic nuclear transcription factors and decreasing lipid peroxidation in the serum and muscle of quail. Curcumin also helps to maintain the antioxidant status of the cells through scavenging free radicals, suppressing oxidative enzymes, and prompting de novo glutathione synthesis [41]. Similarly, the results of Zhai et al. [42] indicated that curcumin compounds could reduce the oxidative injury and lipid metabolism disruption by modifying the cecum microbiota of White Pekin ducklings. Contrarily, the findings of Hosseini-Vashan et al. [43] showed no significant difference in the relative weights of birds offered turmeric. The decreased RBCs and WBCs for chicks produced from eggs exposed to high incubation temperature could be attributed to the adverse effect of heat stress on the physiological status of the birds, consequently affecting oxygen uptake in the eggs for embryos.

The significant decrease in the H/L ratio in chicks produced from treated eggs with curcumin may be due to chick embryos’ improved natural cellular immunity during the incubation phase and consequently after they hatch. The H/L ratio is a known index to measure stress in chicks [29]. These findings also agreed with those of Morita et al. [44], who found a significantly higher H/L ratio for chicks produced from eggs incubated at high incubation temperature (39 °C), while the lower ratio was recorded for chicks incubated at 36 °C.

The decreased glucose level for chicks produced from stressed eggs in the chronic incubation temperature may be due to faster yolk sac absorption. The significant decrease of T3 hormone concentration in the chronic group may be due to depressed activity of the thyrotrophic axis in chicks as reflected by reduced T3 concentration, resulting in functional hypothyroidism. These results agreed with those of Badran et al. [45], who found that T3 concentrations were significantly (*p <* 0.05) lowered for chicks in the chronic group compared with control. Yalçın et al. [46] also found a reduction in T3 hormone concentration in heat acclimated broilers than the control one. The H/L ratio, glucose, and T3 hormone levels for newly-hatched chicks were significantly affected by curcumin treatments (*p <* 0.001). This may be due to the role of curcumin on the immune-stimulating factor in the chick immune system. These findings are in agreement with those of Aamir et al. [47], indicated that the H/L ratio of laying hens supplemented with curcumin under high-temperature conditions was significantly reduced compared with those under normal conditions. Sahin et al. [48] also found that the T3 hormone level was significantly increased in broiler chicks fed 400 mg of chromium/kg diet under high ambient temperature (32 °C).

The decreased body weight for chicks in the chronic group could be attributed to the variability in the initial body weight of the chick, which reflected the final body weight. These findings agreed with those of Abuoghaba [25], who found that the body weight for chicks exposed to chronic temperature (40 °C) was significantly decreased compared with those in the control (37.5 °C).

The significant increase in feed consumption for chicks produced from eggs under normal incubation temperature could be attributed to the normal intestinal development of chicks at hatch [1]. The obtained findings are in agreement with Palo et al. [49], who found that gastrointestinal tract development has a major role in the growth of chick during the early growing period post-hatch.

Regarding the effect of curcumin treatment, it could be noted that body weight at 8 weeks of age and total weight gain was significantly (*p* < 0.05) increased due to curcumin treatment (250, 500, and 1000 mg curcumin/liter distilled water) as compared with the control. However, body weights at the 2nd, 4th, and 6th weeks of age were not affected by curcumin treatments containing 250, 500, or 1000 mg. The beneficial curcumin effects on broiler performance may be due to enhanced amylase secretions, chymotrypsin, trypsin, and lipase enzymes [50]. The improved body weight and weight gain of chickens treated with curcumin may also be due to increased intestinal ecology and morphology by increasing the number of *Lactobacillus* and intestinal villi height of broiler chickens [51]. The results of Rahmani et al. [52] showed that curcumin/nanocurcumin supplementation improved jejunal villus height and villus surface area. The effect of curcumin on increased villus height may be due to increased intestinal epithelial cell turnover, which is associated with activated cell mitosis [53], and then longer villus, which can provide more villus height and enhance nutrient absorption [54]. These results agreed with Reda et al. [55], who found that the body weight of Japanese quails fed diets containing nano-curcumin at 0.1, 0.2, 0.3, 0.4, or 0.5 g/kg significantly increased body weight at 3 and 5 weeks of age. They added that the best body weight for quails enriched with nano-curcumin was at 0.2 or 0.4 g/kg. The higher feed consumption in treated chicks might be due to the well-reported antioxidant, anti-inflammatory, and antibacterial activities [56], or prebiotic-like effects of curcumin [57].

From Table 6 and Table 7, we can observe that the bursa of Fabricius (%) for male and female chickens in the chronic group was significantly decreased compared with the control group. Bursa percentage and weight of spleen were significantly lower in the chronic group, which agrees with other results that indicated that the bursa and spleen in birds are affected by high eggshell temperature [58]. In addition, the significant decrease of the bursa of Fabricius (%) in male and female chickens in the chronic group may be attributed to the negative correlation of bursa histology with high incubation temperature, which resulted in a 7% lower cell density within the follicles of bursa [59]. The broiler chicks that are susceptible to stress present higher O^2^ carrying capacity, blood viscosity, and the number of red blood cells, hematocrit value, and a lower relative heart weight than normal broilers [60]. The results agreed with those of Abuoghaba [25], who found that the bursa of Fabricius (%) for broilers exposed to chronic group (40.0 °C) was significantly decreased compared with those exposed to the control group (37.5 °C). Similarly, Attia et al. [61] found that the spleen and bursa of Fabricius of broiler chicks were significantly affected by the dietary supplementations of curcumin at 0.5, 1, or 2 g/kg diet.

Slaughter weight was significantly increased by the spraying of curcumin (*p* > 0.05) in the current study. Curcumin has revealed antioxidative and antioxidant functions due to its protective effects against oxidative stress [62]. Similarly, the dietary supplementation of curcumin (200 mg/kg diet) significantly increased live body weight in broilers [63], broiler chicken growth performance, and breast yield [64]. In chickens, the hypothalamus–pituitary–gonad axis regulates the development of the reproductive organs, follicular development, follicular maturation, and ovulation [65]. Curcumin significantly enhances the levels of FSH, LH, and estradiol in serum in heat-stressed laying hens, indicating that curcumin supplementation can improve productive performance [36].

The results of Al-Sultan [66] showed that different levels of turmeric did not affect the percentages of heart, liver, and gizzard. The findings of Mehala and Moorthy [67] also showed that the carcass percentage of broiler chickens treated with a 10 g turmeric powder /kg diet was insignificantly affected. These findings agreed with Wang et al. [64] indicated that the addition of 100–300 mg/kg of curcumin had no significant effect on the eviscerated percentage of Wenchang broilers.

## 5. Conclusions

From these results, we can conclude that:(1)During the early embryonic stage, exposure incubated eggs to high temperature (39.0 °C) significantly reduced the hatchability, chick weight at hatch, and percentages of heart, gizzard, RBCs, bursa of Fabricius, and subsequently growth during the growing period.(2)Using curcumin suspension spray on hatching eggs reduced relative water loss, physiological body reaction, and H/L ratio, as well as improved hatchability of fertile eggs, heart, gizzard, spleen, glucose, T3 hormone, body weight, body weight gain, total feed consumption, testes, and ovary.(3)The best hatchability and productive traits were achieved in quails eggs sprayed with 250 mg curcumin/litter to reduce the dangerous effect of high eggshell temperatures during the early incubation phase.

## Figures and Tables

**Table 1 animals-11-03220-t001:** Impact of incubation temperature, curcumin manipulations, and their interaction on relative water loss and embryonic mortality and hatchability.

Trait	Initial Egg Weight (g)	EW-8d (g)	Relative Water Loss (%)	Dead after Piping (%)	Embryonic Mortality Rate (%)	Hatchability	
						Set Eggs (%)	Fertile Eggs (%)
Incubation temperature (IT)							
Control (37.5 °C)	48.85	47.60 ^a^	2.54 ^b^	7.07 ^b^	8.54	80.38	84.39 ^a^
Chronic (39.0 °C)	48.38	46.48 ^b^	3.90 ^a^	8.26 ^a^	9.12	79.11	82.60 ^b^
SEM	0.30	0.29	0.26	0.29	0.56	0.70	0.53
Curcumin manipulations (CM)							
1st (Control)	48.58	46.63	3.99 ^a^	8.26	9.93	77.94	81.81 ^b^
2nd (250 mg)	48.62	47.13	3.07 ^ab^	7.62	9.21	79.67	83.18 ^ab^
3rd (500 mg)	48.58	47.27	2.69 ^b^	7.28	7.87	80.53	84.83 ^a^
4th (1000 mg)	48.68	47.14	3.13 ^ab^	7.49	8.34	80.83	84.17 ^a^
SEM	0.43	0.40	0.36	0.41	0.80	1.00	0.75
Probability							
IT	0.278	0.007	0.004	0.006	0.457	0.207	0.019
CM	0.998	0.690	0.037	0.380	0.282	0.177	0.033
IT × CM	0.979	0.888	0.812	0.975	0.873	0.956	0.929

^a,b^ Means with different superscripts in the same column are significantly different (*p* < 0.05). EW-8d(g), Egg weight-8 day; IT, incubation temperature; CM, curcumin manipulations; SEM, standard error of mean. The values are the average of group data.

**Table 2 animals-11-03220-t002:** Impact of incubation temperature, curcumin manipulations, and their interaction on hatchling quality traits (after 12 h from hatch), and physiological body reactions of newly-hatched chicks.

Trait	Hatchling Quality Traits		Physiological Body Reactions		
	Chick Weight at Hatch (CWAT/g)	Relative Chick Weight (RCW/%)	Chick Length (cm)	Rectal Temperature (RT/°C)	Body Surface Temperature (°C)
Incubation temperature (IT)					
Control (37.5 °C)	35.12 ^a^	71.89	15.95	39.39 ^b^	31.57 ^b^
Chronic (39 °C)	33.70 ^b^	69.69	15.92	39.87 ^a^	32.02 ^a^
SEM	0.23	0.53	0.17	0.14	0.11
Curcumin manipulations (CM)					
1st (Control)	33.60	69.21	15.90	40.29 ^a^	32.72 ^a^
2nd (250 mg)	34.02	69.99	15.96	39.65 ^b^	31.80 ^b^
3rd (500 mg)	34.11	70.37	15.94	39.19 ^b^	31.52 ^c^
4th (1000 mg)	33.89	69.63	15.94	39.40 ^b^	31.12 ^bc^
SEM	0.32	0.75	0.25	0.19	0.15
Probability					
IT	0.020	0.759	0.913	0.015	0.008
CM	0.702	0.727	0.999	0.001	0.001
IT × CM	0.953	0.999	0.998	0.921	0.950

^a,b,c^ Means with different superscripts in the same row are significantly different (*p* < 0.05). IT, incubation temperature; CM, curcumin manipulations; SEM, standard error of mean. The values are the average of group data.

**Table 3 animals-11-03220-t003:** Impact of incubation temperature, curcumin manipulations and their interaction on hatchling organ percentage.

Trait	Heart (%)	Intestine (%)	Gizzard (%)	Liver (%)	Spleen (%)
Incubation temperature (IT)					
Control (37.5 °C)	0.652 ^a^	3.414	2.145 ^a^	2.292	0.085
Chronic (39.0 °C)	0.510 ^b^	3.215	1.914 ^b^	2.198	0.081
SEM	0.046	0.122	0.06	0.08	0.003
Curcumin manipulations (CM)					
1st (Control)	0.441 ^b^	3.057	1.650 ^d^	2.122	0.067 ^b^
2nd (250 mg)	0.575 ^ab^	3.221	2.080 ^c^	2.284	0.086 ^a^
3rd (500 mg)	0.667 ^a^	3.544	2.215 ^a^	2.294	0.091 ^a^
4th (1000 mg)	0.642 ^a^	3.436	2.174 ^b^	2.282	0.088 ^a^
SEM	0.06	0.17	0.04	0.06	0.004
Probability					
IT	0.034	0.256	0.001	0.437	0.428
CM	0.048	0.202	0.001	0.706	0.001
IT × CM	0.953	0.997	0.001	0.996	0.996

^a,b,c,d^ Means with different superscripts in the same row are significantly different (*p* < 0.05). IT, incubation temperature; CM, curcumin manipulations; SEM, standard error of mean. The values are the average of group data.

**Table 4 animals-11-03220-t004:** Impact of incubation temperature, curcumin manipulations and their interaction on hematological parameters, glucose level and T3 hormone concentration in newly-hatched chicks.

Trait	Red Blood Cells (×10^6^)	White Blood Cells (×10^3^)	Hemoglobin (g dL−1)	H/L Ratio	Glucose (mg dL^−1^)	Triiodothyronine (T3/ng/mL)
Incubation temperature (IT)						
Control (37.5 °C)	4.157 ^a^	19.200 ^a^	11.20	0.3207 ^b^	139.42 ^a^	164.49 ^a^
Chronic (39.0 °C)	3.663 ^b^	17.750 ^b^	10.79	0.3780 ^a^	112.15 ^b^	152.71 ^b^
SEM	0.04	0.27	0.25	0.003	0.54	0.32
Curcumin manipulations (CM)						
1st (Control)	3.860	17.90	10.41	0.3585 ^a^	146.14 ^d^	113.75 ^c^
2nd (250 mg)	3.960	18.72	11.12	0.3595 ^a^	160.25 ^c^	125.21 ^b^
3rd (500 mg)	3.913	18.84	11.40	0.3433 ^b^	165.01 ^a^	131.18 ^a^
4th (1000 mg)	3.914	18.40	11.04	0.3362 ^b^	163.02 ^b^	133.02 ^a^
SEM	0.06	0.38	0.36	0.004	0.46	0.76
Probability						
IT	<0.0001	0.0002	0.2553	<0.0001	<0.0001	<0.0001
CM	0.7310	0.2934	0.2582	<0.0001	<0.0001	<0.0001
IT × CM	0.2209	0.0206	0.3783	<0.0001	<0.0001	0.0001

^a,b,c,d^ Means with different superscripts in the same row are significantly different (*p* < 0.05). H/L, Heterophile/lymphocyte ratio; IT, incubation temperature; CM, curcumin manipulations; SEM, standard error of mean. The values are the average of group data.

**Table 5 animals-11-03220-t005:** Impact of incubation temperature, curcumin manipulations and their interaction on productive performance.

Trait	Body Weight (g)/Week					TBWC (g)	DWG (g)	TFC (g)	FCR (g Feed/g Gain)
	One Day	2nd	4th	6th	8th				
Incubation temperature									
Control (37.5 °C)	36.2 ^a^	239.7 ^a^	343.3 ^a^	446.2 ^a^	717.2^a^	680.9 ^a^	12.2 ^a^	1529.2 ^a^	2.25
Chronic (39.0 °C)	34.1 ^b^	232.7 ^b^	332.9 ^b^	436.7 ^b^	650.0 ^b^	615.9 ^b^	10.9 ^b^	1436.7 ^b^	2.33
SEM	0.25	1.06	2.96	3.17	6.32	6.32	0.11	22.71	0.04
Curcumin manipulations (CM)									
1st (Control)	34.9	235.5	332.5	426.7	648. 6 ^b^	613.7 ^b^	10.9 ^b^	1423.8 ^b^	2.33
2nd (250 mg)	35.1	235.9	345.0	453.7	699.5 ^a^	664.4 ^a^	11.7 ^a^	1460.8 ^b^	2.20
3rd (500 mg)	35.6	238.4	342.5	445.0	688.2 ^a^	652.6 ^a^	11.6 ^a^	1475.0 ^b^	2.26
4th (1000 mg)	35.0	235.1	332.5	440.3	698.1 ^a^	663.1 ^a^	11.8 ^a^	1572.2 ^a^	2.38
SEM	0.36	1.49	4.19	4.49	8.95	8.95	0.16	32.1	0.06
Probability									
IT	0.0001	0.002	0.024	0.050	0.0001	0.0001	0.0001	0.011	0.163
CM	0.561	0.432	0.100	0.085	0.003	0.003	0.003	0.029	0.237
IT × CM	0.885	0.186	0.646	0.046	0.059	0.085	0.085	0.075	0.475

^a,b^ Means with different superscripts in the same row are significantly different (*p* < 0.05). DWG (g), daily weight gain; TFC (g), total feed consumption; FCR, feed conversion ratio; IT, incubation temperature; CM, curcumin manipulations; SEM, standard error of mean. The values are the average of group data.

**Table 6 animals-11-03220-t006:** Impact of incubation temperature, curcumin manipulations, and their interaction on carcass characteristics of male chicks.

Trait	LBW (g)	Internal Organ Percentages				Immune Organs		Testes (%)	Eviscerated Carcass (%)
		Heart (%)	Intestine (%)	Gizzard (%)	Liver (%)	Spleen (%)	Bursa (%)		
Incubation temperature (IT)									
Control (37.5 °C)	801.0 ^a^	0.606	7.27	3.06	2.82 ^a^	0.339	0.246 ^a^	0.208	63.44
Chronic (39.0 °C)	693.6 ^b^	0.611	6.90	2.91	2.69 ^b^	0.335	0.223 ^b^	0.206	61.79
SEM	14.24	0.02	0.28	0.09	0.07	0.01	0.01	0.009	1.19
Curcumin manipulations (CM)									
1st (Control)	709.1 ^b^	0.568	6.44 ^b^	2.74 ^b^	2.55 ^b^	0.306	0.224	0.153 ^c^	60.64
2nd (250 mg)	746.3 ^ab^	0.612	7.38 ^ab^	3.03 ^ab^	2.82 ^ab^	0.346	0.238	0.215 ^b^	63.18
3rd (500 mg)	781.5 ^a^	0.637	7.86 ^a^	3.28 ^a^	2.94 ^a^	0.352	0.243	0.253 ^a^	64.83
4th (1000 mg)	752.3 ^ab^	0.619	6.65 ^ab^	2.88 ^ab^	2.71 ^ab^	0.345	0.233	0.206 ^b^	61.81
SEM	20.14	0.03	0.40	0.13	0.09	0.02	0.01	0.01	1.68
Probability									
IT	<0.0001	0.883	0.361	0.279	0.052	0.769	0.049	0.856	0.338
CM	0.013	0.473	0.038	0.050	0.049	0.247	0.646	0.001	0.356
IT × CM	0.010	0.257	0.279	0.218	0.199	0.016	0.083	0.047	0.822

^a,b,c^ Means with different superscripts in the same row are significantly different (*p* < 0.05). LBW (g), life body weight; IT, incubation temperature; CM, curcumin manipulations; SEM, standard error of mean. The values are the average of group data.

**Table 7 animals-11-03220-t007:** Impact of incubation temperature, curcumin manipulations, and their interaction on carcass traits of female chicks.

Traits	LBW (g)	Internal Organ Percentages				Immune Organs		Ovary (%)	Eviscerated Carcass (%)
		Heart (%)	Intestine (%)	Gizzard (%)	Liver (%)	Spleen (%)	Bursa (%)		
Incubation temperature (IT)									
Control (37.5 °C)	647.8 ^a^	0.556	6.58 ^a^	2.76	2.65 ^a^	0.333	0.230 ^a^	0.066	61.7
Chronic (39.0 °C)	613.9 ^b^	0.548	5.78 ^b^	2.52	2.38 ^b^	0.321	0.205 ^b^	0.061	59.0
SEM	7.56	0.021	0.12	0.097	0.093	0.010	0.006	0.004	1.17
Curcumin manipulations (CM)									
1st (Control)	605.2 ^b^	0.531	5.92	2.33 ^b^	2.25	0.307	0.209	0.054 ^b^	62.2
2nd (250 mg)	643.0 ^a^	0.561	6.21	2.78 ^a^	2.60	0.331	0.222	0.075 ^a^	61.3
3rd (500 mg)	651.5 ^a^	0.562	6.34	2.84 ^a^	2.68	0.346	0.230	0.067 ^ab^	59. 6
4th (1000 mg)	623. 7 ^ab^	0.554	6.24	2.62 ^ab^	2.53	0.323	0.218	0.058 ^ab^	58.5
SEM	10.7	0.03	0.17	0.14	0.13	0.015	0.008	0.005	1.65
Probability									
IT	0.006	0.787	0.001	0.094	0.055	0.438	0.018	0.379	0.127
CM	0.033	0.863	0.374	0.047	0.163	0.341	0.317	0.047	0.411
IT × CM	0.025	0.038	0.675	0.200	0.912	0.057	0.800	0.001	0.006

^a,b^ Means with different superscripts in the same row are significantly different (*p* < 0.05). LBW (g), life body weight; IT, incubation temperature; CM, curcumin manipulations; SEM, standard error of mean. The values are the average of group data.

## Data Availability

All data sets obtained and analyzed during the experiment are available on fair request from the respective author.

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
