# Peer review of "Impact of Treating Hatching Eggs with Curcumin after Exposure to Thermal Stress on Embryonic Development, Hatchability, Physiological Body Reactions, and Hormonal Profiles of Dokki-4 Chickens"

_animals, 2021, doi:10.3390/ani11113220_

Round 1

Reviewer 1 Report

This paper studied the potential of using curcumin suspension spray as an alternative way to reduce the negative impact of high temperature on chick embryo developments and hatchability of Dokki chickens. The work seems to be carried out with caution and care and positive results were obtained. The overall English writing is also good. In general, I think this paper can be published in Animals (MDPI). Yet, I urge the authors to consider the following suggestions to improve this quality of this manuscript.

  • The authors need to provide more discussions on the novelty of this paper. In my opinion, I think the temperature effect on the hatchability of chicken egg as well as the methods to deal with such effects should have been done by other investigators. The authors should provide more justifications of the novelty of their work as well as more literature surveys of others’ work.
  • The authors pointed out that using curcumin to eliminate the negative temperature effect on hatchability is more cost-effective in developing country. Yet, I think curcumin is not a very cheap chemical. Using curcumin in a farm will add additional cost. Is it really feasibly? Can the authors provide more justifications on the actual cost?

Author Response

Corrections according to reviewer comments   

Reviewer 1

Comments and suggestions for authors

This paper studied the potential of using curcumin suspension spray as an alternative way to reduce the negative impact of high temperature on chick embryo developments and hatchability of Dokki chickens. The work seems to be carried out with caution and care and positive results were obtained. The overall English writing is also good. In general, I think this paper can be published in Animals (MDPI). Yet, I urge the authors to consider the following suggestions to improve this quality of this manuscript.

Answer:

  • We provided sufficient justification for the novelty of their work. (line 341-343 in revised manuscript).
  • Generally, spraying hatching eggs with curcumin is a new point in the field of hatching chicken eggs.

  • The authors pointed out that using curcumin to eliminate the negative temperature effect on hatchability is more cost-effective in developing country. Yet, I think curcumin is not a very cheap chemical. Using curcumin in a farm will add additional cost. Is it really feasibly? Can the authors provide more justifications on the actual cost?

Answer:

We pointed out that the use of curcumin to get rid of the negative effect of temperature on hatchability is more cost-effective in developing countries because of its cheap price. The price of one gram of curcumin is about 1 pound in Egypt, which is equivalent to about 0.16 dollars. Through the achieved results, we find that it is economically inexpensive.

The authors are thankful for the insightful review of the present article.

Reviewer 2 Report

Manuscript ID: animals-1396739

Title: Impact of treating hatching eggs with curcumin after exposure to thermal stress on embryonic development, hatchability, physiological body reactions and hormonal profiles of Dokki-4 chickens.

The idea is good and the paper contains useful information. The manuscript is interesting. But, there are some adjustments (n=54) should be considered before publication.

(1) Line 26: Add "the" before "control"

(2) Line 72: Put this sentence "Curcumin is used as a food  ……….." in new paragraph

(3) There is repetition in these sentences (lines 72-74, lines 77-78 and lines 83-84)

(4) Line 80: This reference (16, Zhang et al., 2015) did not use quail, but used broilers

(5) Line 87: What does this term mean "HATs"?

(6) The goal at the end of the introduction section is to ignore the topic of exposure to thermal stress

(7) Line 114: Change " 6 to 8 day " to " 6th to 8th day"

(8) Line 114: why??? from 6 to 8 day of incubation, which eggs were daily exposed to 39.0°C for 3 hours from  12 to 3 PM ………. Whly from 6 to 8 day??

(9) Line 141: Change " Piping " to "piping"

(10) Line 172: Why " ) "?

(11) Lines 174-176: To what reference did you base these requirements on (CP and ME)? It must be an official reference.

(12) Lines 177-180: To what reference did you base these requirements on (light)? It must be an official reference.

(13) Line 206: Change " decreases" to " decreased"

(14) Line 227: " rectal (RT/°C)" What does the letter T mean here?

(15) Line 241, 266, 292, 396: Change " lower" to " lowered"

(16) Lines 255, 249and 380: Why repetition? (×106)  (×103)

(17) Line 251 and 267-268: "was significantly (p < 267 0.01) higher" change " higher " to "increased" and correct the rest sentence

(18) Line 274-275: Add "was" before " significantly increased "

(19) In Table 5: " FCR (g feed/g meat) ". meat is not correct.

(20) Line 318: Change " supplement " to " supplemented"

(21) Line 321: " The authors " who are they?

(22) Line 325: Add "was" before " significantly (p = 0.01) decreased "

(23) Lines 330-333: Rephrease this sentence

(24) Line 359: Add "were" before " significantly decreased "

(25) Line 360-368: This interpretation indicates the presence of pathological conditions. Is low weight in birds of your experiment  indicates the presence of satisfactory cases?. Reconsider this interpretation.

(26) Line 405: Change " increase " to " increased"

(27) Line 409: Change " bodyweight " to " body weight "

(28) Line 410: Change " chick " to " chicks"

(29) Line 410: Add "was" before " significantly decreased "

(30) Line 419: Change " Curcumin" to "curcumin"

(31) Line 425: " Lactobacillus " should be italic

(32) Line 447: " Abuoghaba [22] " or " Abuoghaba [25] " is correct?

(33) Line 475: These abbreviations "BW, BWG, TFC" are not used in any of the previous sentences.  Why are you using it here?

(34) Line 476: You did not mention that this level is the best within the text in the results or Abstract. Why did you choose it here in the conclusion that it is the best?

(35) Line 487: the author reportd that "This research received no external funding." But in line  493 reported that "This research was realized from statutory research funds assigned by Sohag University, Egypt. " There is inconsistency. Which is correct?

(36) Lines 488-489: The study was approved by the Institutional Animal Care and Use Committee at Sohag University, Egypt (6-1-2020). But in line 106 approval number was (no. 13/2020). There is inconsistency. Which is correct?

(37) Lines 500, 618, 620: " . . " Delete one

 (38) Reference 12 in line 525 is the same as Reference 19 in line 542. Please Delete one, and Check the arrangement carefully

(39) Reference 50 in line 615 is the same as Reference 60 in line 641. Please Delete one and Check the arrangement carefully.

(40) Reference 13 in line 528:  This reference is missing the title

(41) Line 533: "2012" should be blod

(42) Line 541: "2014" should be blod

(43) Line 551, 556: "2009" should be blod

(44) Line 551, 556: " Bioorg. Med. Chem." should be italic

(45) Line 587: " Arch. Toxicol." should be italic

(46) Line 587: "2010" should be blod

(47) Line 590: " Poult. Sci. " should be italic

(48) Line 590: "2020" should be blod

(49) Line 617: "2018" should be blod

(50) Line 617: Check this information "J. Appl. Anim. Res., 2018, 46, 200–209"

(51) Line 619: "2002" should be blod

(52) Line 628: "2009" should be blod

(53) Line 642: " Italian J. Anim. Sci." should be italic

(54) Line 654: Change "Aloevera" to " Aloe vera"

Author Response

Corrections according to reviewer comments   

Reviewer 2

Comments and Suggestions for Authors

Manuscript ID: animals-1396739

Title: Impact of treating hatching eggs with curcumin after exposure to thermal stress on embryonic development, hatchability, physiological body reactions and hormonal profiles of Dokki-4 chickens.

The idea is good and the paper contains useful information. The manuscript is interesting. But, there are some adjustments (n=54) should be considered before publication.

(1) Line 26: Add "the" before "control"

Answer: The correction was done (line 26 in revised manuscript). 

(2) Line 72: Put this sentence "Curcumin is used as a food ……….." in new paragraph

Answer: The correction was done (line 73 in revised manuscript).

(3) There is repetition in these sentences (lines 72-74, lines 77-78 and lines 83-84)

Answer: in line 73-83  in revised manuscript

Note: The new references were added to references:

(4) Line 80: This reference (16, Zhang et al., 2015) did not use quail, but used broilers

Answer: we add the broilers after quails to be “quails and broilers” line 80 in revised manuscript

(5) Line 87: What does this term mean "HATs"?

Answer: histone acetyltransferases(HATs) line 86 in revised manuscript

(6) The goal at the end of the introduction section is to ignore the topic of exposure to thermal stress

Correction: Aim of this experiment was to evaluate the impact of spraying hatching eggs after exposure to thermal stress (39°C) (lines 98-99 in revised manuscript)

(7) Line 114: Change "6 to 8 day" to "6th to 8th day"

Correction: The correction was done line 112 in revised manuscript

(8) Line 114: why??? from 6 to 8 day of incubation, which eggs were daily exposed to 39.0°C for 3 hours from  12 to 3 PM ………. Whly from 6 to 8 day??

Answer:

  • Chick embryos are more sensitive to heat during the first third of incubation (1-8 days), due to the formation and development of their thermoregulatory system, while with the increase in the development and growth of embryos, chick embryos become more affected by the rate of ventilation to keep up with their oxygen needs.
  • According to French (1997), eggs will absorb heat from the surrounding air during the first half of incubation due to embryo temperature being slightly lower than incubator temperature, but embryos must lose heat during the second half of incubation as their metabolic rate and heat production increase.

(9) Line 141: Change “Piping “to "piping"

Correction: The correction was done line 138 in revised manuscript

(10) Line 172: Why " ) "?

Answer: “) “was deleted 

(11) Lines 174-176: To what reference did you base these requirements on (CP and ME)? It must be an official reference.

Answer: According to NRC, [30] line 175 in revised manuscript

(12) Lines 177-180: To what reference did you base these requirements on (light)? It must be an official reference.

Answer: The reference was added in line 177 in revised manuscript

- Note: There are 3 abbreviated terms (relative water loss, embryonic mortality and dead after piping) written in a non-abbreviated form (line 198, 201, 202-203 in revised manuscript)

(13) Line 206: Change “decreases" to “decreased"

Correction: The correction was done line 206 in revised manuscript

(14) Line 227: “rectal (RT/°C)" What does the letter T mean here?

Answer: T mean temperature and changed in full form to be “Temperature” line 227 in revised manuscript and in Table 2

(15) Line 241, 266, 292, 396: Change " lower" to " lowered"

Correction: The correction was done line 240, 265, 291, 407 in revised manuscript

(16) Lines 255, 249and 380: Why repetition? (×106)  (×103)

Answer: The repetition was deleted except line 248

(17) Line 251 and 267-268: "was significantly (< 267 0.01) higher" change " higher " to "increased" and correct the rest sentence

Correction: The correction was done line 249, 267  in revised manuscript

(18) Line 274-275: Add "was" before" significantly increased”

Answer: The correction was done line 273  in revised manuscript

(19) In Table 5: “FCR (g feed/g meat) ". meat is not correct.

Correction: It changed to be FCR (g feed/ g gain)

(20) Line 318: Change “supplement “to “supplemented"

Correction: The correction was done in line 317 in revised manuscript

(21) Line 321: “The authors “who are they?

Correction: The correction was done in line 320 in revised manuscript

(22) Line 325: Add "was" before " significantly (= 0.01) decreased”

Correction: The correction was done in line 324 in revised manuscript

(23) Lines 330-333: Rephrease this sentence

Answer: The sentence rewrite to be “The significant (p = 0.01) increase in the dead after piping for chick embryos in the chronic group could be attributed to the deleterious effects of heat stress, which negatively affect the pulmonary vascular capacity leading to increasing the oxygen metabolic demand [38]”. In line 329-332 in revised manuscript

In line 339: of fertile eggs was added

(24) Line 359: Add "were" before “significantly decreased”

Correction: The correction was done in line 361 in revised manuscript

(25) Line 360-368: This interpretation indicates the presence of pathological conditions. Is low weight in birds of your experiment  indicates the presence of satisfactory cases?. Reconsider this interpretation.

Answer: in line 364-372 in revised manuscript

  • This explanation has been rewritten
  • Low weight in birds from my experience does not indicate the presence of pathological conditions, but I mean that the exposure of birds to heat stress may be susceptible in the future to exposure to diseases, although no diseases were recorded during the experiment period.

(26) Line 405: Change " increase " to " increased"

Correction: The correction was done in line 416  in revised manuscript

(27) Line 409: Change " bodyweight " to " body weight "

Correction: The correction was done in line 420  in revised manuscript

(28) Line 410: Change " chick " to " chicks"

Correction: The correction was done in line 421 in revised manuscript

(29) Line 410: Add "was" before " significantly decreased "

Correction: The correction was done in line 421 in revised manuscript

(30) Line 419: Change " Curcumin" to "curcumin"

Correction: The correction was done in line 430  in revised manuscript

(31) Line 425: " Lactobacillus " should be italic

Correction: The correction was done in line 436 in revised manuscript

(32) Line 447: " Abuoghaba [22] " or " Abuoghaba [25] " is correct?

Correction: The correction was done in line 458  in revised manuscript

(33) Line 475: These abbreviations "BW, BWG, TFC" are not used in any of the previous sentences.  Why are you using it here?

Correction: The correction was done in line 486-487 in revised manuscript

(34) Line 476: You did not mention that this level is the best within the text in the results or Abstract. Why did you choose it here in the conclusion that it is the best?

Answer: The best hatchability and productive traits were achieved in quails eggs sprayed with 250 mg curcumin/litter to reduce the dangerous effect of high eggshell temperatures during the early incubation phase. In line 488-490  in revised manuscript

(35) Line 487: the author reportd that "This research received no external funding." But in line 493 reported that "This research was realized from statutory research funds assigned by Sohag University, Egypt." There is inconsistency. Which is correct?

Answer: The authors are very grateful to conduct the research at the Poultry Production Department, Faculty of Agriculture, Sohag University, Egypt. In line 504-505  in revised manuscript

(36) Lines 488-489: The study was approved by the Institutional Animal Care and Use Committee at Sohag University, Egypt (6-1-2020). But in line 106 approval number was (no. 13/2020). There is inconsistency. Which is correct?

Answer: Date of approval to conduct the search (6-1-2020) in line 500 in revised manuscript

(37) Lines 500, 618, 620: " . . " Delete one

Answer: The correction was done 

(38) Reference 12 in line 525 is the same as Reference 19 in line 542. Please Delete one, and Check the arrangement carefully

  • Galli, G.M.; Da Silva, A.S.; Biazus, A.H.; Reis, J.H.; Boiago, M.M.; Topazio, J.P.; Santos, C.G. Feed addition of curcumin to laying hens showed anticoccidial effect, and improved egg quality and animal health. Vet. Sci. 2018, 118, 101–106. https://doi.org/10.1016/j.rvsc.2018.01.022

Answer: The reference 19 in line 542 was deleted

(39) Reference 50 in line 615 is the same as Reference 60 in line 641. Please Delete one and Check the arrangement carefully. in lines 637-658  in revised manuscript

Answer: There is a difference between both references, there is a difference in the studied traits, and there is no similarity in the studied traits

Rahmani, M.; Golian, A.; Kermanshahi, H.; Bassami, R. M. Effects of curcumin or nanocurcumin on blood biochemical parameters, intestinal morphology and microbial population of broiler chickens reared under normal and cold stress conditions. J. Appl. Anim. Res., 2018, 46, 200–209. https://doi.org/10.1080/09712119.2017.1284077

Rahmani, M.; Golian, A.; Kermanshahi, H.; Bassami, M.R. Effects of curcumin and nanocurcumin on growth performance, blood gas indices and ascites mortalities of broiler chickens reared under normal and cold stress conditions. Italian J. Anim. Sci. 2017, 16, 438-446. https://doi.org/10.1080/1828051X.2017.1290510  

(40) Reference 13 in line 528: This reference is missing the title

Answer: The reference title was corrected in line 545-546  in revised manuscript

(41) Line 533: "2012" should be bold

Answer: The correction was done in line 548 in revised manuscript

(42) Line 541: "2004" should be bold

Answer: The correction was done in line 558  in revised manuscript

(43) Line 551, 556: "2009" should be bold

Answer: The correction was done in line 567  in revised manuscript

(44) Line 551, 556: " Bioorg. Med. Chem." should be italic

Answer: The correction was done in line 567 in revised manuscript

(45) Line 587: " Arch. Toxicol." should be italic

Answer: The correction was done in line 567  in revised manuscript

(46) Line 587: "2010" should be bold

Answer: The correction was done in line 606 in revised manuscript

(47) Line 590: " Poult. Sci. " should be italic

Answer: The correction was done in line 608 in revised manuscript

(48) Line 590: "2020" should be bold

Answer: The correction was done in line 609  in revised manuscript

(49) Line 617: "2018" should be bold

Answer: The correction was done in line 636 in revised manuscript

(50) Line 617: Check this information "J. Appl. Anim. Res., 2018, 46, 200–209"

Answer: The correction was done in line 636 in revised manuscript

(51) Line 619: "2002" should be bold

Answer: The correction was done in line 338 in revised manuscript

(52) Line 628: "2009" should be bold

Answer: The correction was done in line 647  in revised manuscript

(53) Line 642: " Italian J. Anim. Sci." should be italic

Answer: The correction was done  in line 661 in revised manuscript

(54) Line 654: Change "Aloevera" to " Aloe vera"

Answer: The correction was done in line 673  in revised manuscript

The authors are thankful for the insightful review of the present article.

Reviewer 3 Report

Simple summary: no comment

Abstract: no comment

Introduction: It needs to be more elaborate about effect of curcumin on methylation.

Material and methods: Line 110 sounds confusing, if possible, rewrite.

Line 168, mentioned commercial kit, please mention kit name and company name.

Result:

Please add the abbreviation of SEM.

Table 1: I am no statistician but significant difference does not look very different.

Table 2: CWAT, RCW, full form.

Discussion: In whole experiment there are some significant differences, but mostly not. But in discussion it is not clear how curcumin here is helping to improve thermal stress and other factors as mentioned in title. It is not clear.

Conclusion: Not very clear, may be need to rewrite.

Author Response

REVIEWER 3

Comments and Suggestions for Authors

Simple summary: no comment

Abstract: no comment

Introduction: It needs to be more elaborate about effect of curcumin on methylation.

Answer: This sentence was added to clear curcumin methylationMoreover, curcumin modulates DNA methylation through inhibiting DNA methyltransferases (DNMTs) [23]”.

Material and methods:

Line 110 sounds confusing, if possible, rewrite.

Correction: This sentence changed to be “In this study, 720 Dokki-4 fertile eggs (47±2g) were collected (3 times/day) from hen breeder flock at 46 weeks of age” in line 107-108  in revised manuscript

168, mentioned commercial kit, please mention kit name and company name.

Answer: Plasma triiodothyronine (T3) was determined by enzyme-linked immunosorbent assay (ELISA) kit (International Reagents Corporation, Kobe, Japan). in line 164-165 in revised manuscript

Result:

Please add the abbreviation of SEM.

Answer: SEM, Standard error of mean in line 211 in revised manuscript

Table 1: I am no statistician but significant difference does not look very different.

Answer: After revision all data, all data in Table 1 analysis is true

Table 2: CWAT, RCW, full form

Correction: CWAT, Chick weight at hatch; RCW, Relative chick weight; T, temperature as well as RWL, relative water loss in conclusion

Discussion: In whole experiment there are some significant differences, but mostly not. But in discussion it is not clear how curcumin here is helping to improve thermal stress and other factors as mentioned in title. It is not clear.

Answer: The discussion for mentioned traits in the title was written in lines 329-332, 342-345, 364-372, 375-382

Conclusion: Not very clear, may be need to rewrite.

Correction: the conclusion section was rewrite to be: in lines 480-490

  • During early embryonic stage, exposure incubated eggs to high temperature (39.0 ºC) significantly reduced the hatchability, chick weight at hatch, and percentages of heart, gizzard, RBCs, bursa of Fabricius and subsequently growth during the growing period. 
  • Using curcumin suspension spray in hatching eggs reduced relative water loss, physiological body reaction, and H/L ratio as well as improved hatchability of fertile eggs, heart, gizzard, spleen, glucose, T3 hormone, body weight, body weight gain, total feed consumption, testes, and ovary. 
  • The best hatchability and productive traits were achieved in quails eggs sprayed with 250 mg curcumin/litter to reduce the dangerous effect of high eggshell temperatures during the early incubation phase.

The authors are thankful for the insightful review of the present article.

Round 2

Reviewer 2 Report

I found all the required comments were checked and revised carefully by the authors. So, I recommend this manuscript for publication in Animals

Reviewer 3 Report

I Have gone through it, and no further comment.

This manuscript is a resubmission of an earlier submission. The following is a list of the peer review reports and author responses from that submission.

Round 1

Reviewer 1 Report

The manuscript by Abuoghaba and others addressed the protective effect of curcumin on heat stress during chick embryogenesis. 

Curcumin is a widely used natural product of which function has been extensively studied at both physiological and molecular levels. 

The manuscript failed to summarize previous studies on the effect of curcumin on heat stress during chick embryogenesis. The authors mentioned some such studies at the discussion, but it should be clearly defined what has been discovered and what remains to be uncovered. The manuscript needs to declare the major undetermined agendas that they will address in their study.

- Answer: No previous studies on the effect of curcumin on heat stress during chick embryogenesis, but many studied only evaluated the effect of curcumin in the diet and in ovo-injection.

Comment: The authors need to state this in the introduction.  

The manuscript also failed to explicitly declare the rationale for the experimental design. For example, what is the rationale for applying curcumin after the heat stress? Why not including pretreatment of curcumin before heat stress?

- Answer: The study aimed at the possibility of using curcumin as an anti- heat stress in addition to being an anti-oxidant in poultry production.

Answer: The rationale for the application of curcumin after heat stress is to know whether it can be used as an anti-heat stress as a natural product used to improve immunity and as an anti-oxidant. Curcumin was treated after exposure to heat stress to know its effect in reducing the heat burden, especially in the summer for young breeders.
The effect of curcumin treatment before heat stress or at the same time will be studied in later studies

Comment: It is clear that heat stress will hamper chick development. Treating curcumin after the heat stress will allow one to observe if curcumin facilitates the recovery process from the heat stress. It is unclear how chicken eggs recover from heat stress. The authors argue that curcumin improves immunity and anti-oxidation function, but they need to present data to support this. As the authors stated earlier, there is no study on the effect of curcumin during heat-stressed chick embryogenesis. In contrast, treating eggs with curcumin before or simultaneously with the heat stress will allow one to correlate curcumin with heat stress. If the aim was to determine if curcumin has an effect to reduce heat burden in the summer, the authors need to explicitly state that is the rationale: Temperature control is not affordable in developing countries would be a good rationale for the study design.  

Various developmental markers showed that chicks treated with curcumin displayed improved development. The manuscript, however, only speculated that curcumin may facilitated chicks recovering from heat stress. Attenuating oxidative stress was mentioned by quoting other studies, but they didn't attempt to obtain any evidence to support their speculation. 

- Answer: Through the study, it was shown the role of curcumin in reducing heat stress, and related productivity measures were taken according to the material capabilities available in Egypt.

Comment: The authors basically did not answer this point. It is possibly beyond the scope of this study, but some suggestions about the mechanism of curcumin-mediated heat stress recovery could be made.  

Stress will antagonize effective development and thus lower various physiological read-outs. Tumeric has been know to help animals to cope better with various stress. The authors suggested that the addition of curcumin would improve the quality of the chicks, but they did not quantitatively compare the merit over the expense of curcumin treatment.

- Answer: It is clear from the different tables that the spraying curcumin improved the performance of birds and reduced the heat stress on them,

and the study recommends the use of curcumin at a level of 500 mg / liter of water in hatching

Comment: The authors are able to calculate the cost of spraying curcumin during the chick embryo development together with the involved labor and compare the gain of improved chick quality in dollar amount. This simple calculation will make the authors' argument much more convincing.    

The manuscript contains too many grammatical errors, typos, and oversight. For example, sentences of lines 42 to 51 are copied from some guidelines and none of these authors noticed this blunder. The authors used both Curcumin and Cur randomly. The authors should read their manuscript seriously and consult a native English-speaking professional before they submit it to a journal.   

- Answer: “Cur” in the manuscript changed to be curcumin

For example, sentences of lines 42 to 51 are copied from some guidelines and none of these authors noticed this blunder.

- Answer: The plagiarism rate was verified in lines 42-51, and it was found that the citation rate was very small

Comment: The original text in lines 42-51 is as follows:  Introduction 42 The introduction should briefly place the study in a broad context and highlight why 43 it is important. It should define the purpose of the work and its significance. The current 44 state of the research field should be carefully reviewed and key publications cited. Please 45 highlight controversial and diverging hypotheses when necessary. Finally, briefly men- 46 tion the main aim of the work and highlight the principal conclusions. As far as possible, 47 please keep the introduction comprehensible to scientists outside your particular field of 48 research. References should be numbered in order of appearance and indicated by a nu- 49 meral or numerals in square brackets—e.g., [1] or [2,3], or [4–6]. See the end of the docu- 50 ment for further details on references.

The authors did not realize this blunder or what? It is not about plagiarism.  

The manuscript still contains multiple grammatical errors and typos. They should be fixed at this second round and the authors should get editorial service from a professional.  One such example is: “The eggshell surfaces were cleaning with 100% ethanol … “ line 121.  It should be “cleaned”.

Reviewer 2 Report

Although this work was conduced well by the authors, and the results showed benefit effects of spring with Cur, the Discussion section still lacks scientific statement after revision. The authors emphasize the significance of this method for developing countries, while they neglected the discussion of the result compound logic. Effects of spraying need reasonable explanations, previous in vivo studies can not support the results in this study, because the mechanisms must be different between spraying and in vivo. Therefore, I do not think this work can reach the advance of Animals.